# New Approaches to Tackling Intractable Issues in Infectious Disease

**DOI:** 10.3390/microorganisms12030421

**Published:** 2024-02-20

**Authors:** Paul Barrow

**Affiliations:** School of Veterinary Medicine, University of Surrey, Daphne Jackson Road, Guildford, Surrey GU2 7AL, UK; paul.barrow@surrey.ac.uk

**Keywords:** *Salmonella*, carriage, immune modulation, antimicrobial resistance, AMR, bacteriophages, plasmids, viruses, parasites

## Abstract

Despite major progress in the last several decades in reducing the public and animal health burden of infectious disease a number of issues remain to be resolved and which have thus far been regarded as intractable. These include (i) the persistent carrier state in individuals convalescent from typhoid and typhoid-like infections, (ii) the increasing prevalence of multi-antibiotic resistance in enteric pathogens, much of which is mediated by self-transmissible plasmids, and (iii) parasite infections which are difficult to control by vaccination and where resistance to chemotherapeutics is also increasing. The author describes very recent work carried out by his group to look at resolving these problems in new and imaginative ways.

## 1. Introduction

Despite the success over the last several decades in reducing the burden of infectious diseases globally, in part from national but also international efforts, progress has been faster in some areas than in others and there remain a number of outstanding persistent issues that require resolution.

Ignorance remains of aspects of the infection biology of some key pathogens where antimicrobial resistance (AMR) is also an increasing problem, such as *Salmonella* Typhi, the causative agent of human typhoid, together with related bacteria where convalescence is associated with persistent shedding and where the infection biology remains poorly understood. Typhoid remains endemic in many countries in South Asia and Latin America and the livestock pathogens, which produce a similar typhoid-like disease, *S.* Gallinarum/Pullorum (poultry), *S*. Dublin (cattle) and *S*. Abortus-ovis (sheep), remain global problems for the agricultural economy in different parts of the world, again with AMR being an increasing problem.

Respiratory and gastrointestinal infections, especially in children, remain major causes of morbidity and mortality in many developing countries, in many cases caused by mixed infections which are more difficult to treat. Vaccines are frequently available for the major pathogens but may not be accessible and are anyway less effective in mixed infections. The consequence can be over-use of chemotherapy which contributes to the global problem of AMR [1]. AMR is the result of use, over-use (regulated and unregulated) and abuse of chemotherapy in human and livestock medicine and as a result of continuing use for growth promotion purposes in many parts of the world, although this is now banned in Europe and the US.

Resistance to chemotherapeutic agents is also an increasing problem in parasite infections in human and veterinary medicine, with *Plasmodium falciparum* (malaria in man) now resistant to most previously effective agents [2]. For many parasite infections, vaccination is not an option and novel approaches to controlling these infectious agents should be explored.

It is thus timely to explore how some of these persistent problems might be addressed through new modes of thought and technologies.

## 2. Immunomodulation of Persistent Typhoid-like *Salmonella* Infections

### 2.1. Infection Biology of Salmonella Infections

Most of the more than 2500 serotypes for *Salmonella enterica* are associated with gastrointestinal diseases in man and livestock. However, a small number of serovars, including *S. enterica* serovar Typhi (*S*. Typhi), *S*. Gallinarum, *S*. Pullorum, *S*. Dublin, *S*. Choleraesuis and *S*. Abortusovis/equi, are adapted to a narrow range of host species and generally produce severe, typhoid-like disease, sometimes with high mortality [3,4]. Although *S*. Typhimurium and *S*. Enteritidis (SE) are the serovars most frequently associated with food-poisoning, with infection these also produce characteristic typhoid experimentally in mice [3]. One of the features of the infection produced by the typhoid serovars is disease-free persistent infections in a proportion of convalescents also observed in experimental infection involving macrophages in lymphoid tissues ([5,6,7], Table 1). This results in localization in the gall bladder, liver and spleen, leading to faecal shedding by carriers for long periods and, in some cases, many years (*S.* Typhi in man and *S.* Dublin in cattle) [1,8,9,10] or localization in the reproductive tract leading either to abortion (*S.* Dublin, *S*. Abortusovis in sheep) or vertical transmission through hatching eggs to progeny (*S.* Pullorum and *S*. Gallinarum), a key aspect of the epidemiology [10]. *S*. Pullorum is a good and natural model of the persistent infection shown by these serovars [7].

The serovars that produce gastrointestinal infections generally stimulate strong immunological and, in many cases, clearing responses [11]. Studies on murine typhoid with *S.* Typhimurium have indicated the critical role of CD4^+^ Th1 lymphocytes in controlling salmonellosis [12]. Clearance of infection by *S*. Typhimurium and *S*. Enteritidis from the intestinal tract of infected chickens was also shown to be due to a Th1-dominated response involving increased expression of IFN-γ mRNA in the gut and deeper tissues and generation of IL-12 and IL-18 early in infection [13,14,15]. 

### 2.2. The Biology of Persistent Infection (Carrier State)

Persistent infection produced by the typhoid-like serovars has long been poorly understood. Microbiological studies on *S*. Pullorum infection in young chickens, which had convalesced from a mild infection, showed that the bacteria persist within splenic macrophages [8] in small number in both sexes. In females, this was interrupted by reduced T cell activity at onset of lay (sexual maturity) not seen in males. Persistence occurs despite very high levels of circulating specific IgG [10]. Comparison of the immune response with a taxonomically closely related serovar *S*. Enteritidis indicated that the latter induced a typical Th1/Th17-type response with high levels of Th1-associated chemokines IFNγ, IL-12α, IL-17F and IL-18 and low levels of Th2-associated chemokines (IL-4 and IL-13). *S.* Pullorum induced the reverse picture of chemokine induction (high Th2-associated cytokines and low Th1 cytokines), which is highly likely to result in poor cell-mediated clearance from the tissues but with high levels of circulating specific IgG. There was little evidence of clonal anergy or immune suppression induced by *S*. Pullorum [15].

That the background genetics of the host also contributes to the persistent carrier phenotype is indicated by the fact that persistence of *S.* Typhimurium in mice occurs in the Slc11a1+/+ (*Salmonella* resistant) phenotype. Similarly, *S.* Gallinarum shows persistence in chickens but only in the resistant *Sal*1^R^ phenotype. In both host species infection of individual animals expressing a disease-susceptible phenotype results in acute disease and death [5,16,17].

The picture in Slc11a1+/+ mice infected with *S.* Typhimurium is similar with persistence in anti-inflammatory splenic M2 macrophages for many months activated by Th2 cytokines. It is thought that the bacteria reprogramme M1 macrophages to a M2 phenotype involving the bacterial protein SteE, which is also found in a high proportion of *S.* Pullorum and *S*. Gallinarum strains (Berchieri and Benevenides, unpublished results). M1 and M2 macrophages also show differences in physiology with the M1 macrophages leading to production of NO typical of bactericidal effects in chronic bacterial infections. Transcripts in carrier mice resemble those of humans carrying *S*. Typhi although it was considered that immune suppression was also involved [18,19,20].

Very little is known about the immune response to the livestock serovars *S*. Dublin and *S*. Abortus-ovis, although lower levels of IL-12 and TNFα were found in calves infected with *S*. Dublin [21]. In *S*. Typhi, there is an indication that convalescents carriers can be split into two classes, one in which the cytokine pattern resembles that of post-acute patients while the other shows reduced lymphocyte numbers and CD8+ cytotoxic T lymphocyte transcripts [22].

### 2.3. Alternative Strategy to Reduction in the Carrier State

There is now increasing interest in remodulating the host immune response away from that driven by pathogens to one which benefits the host. In human leprosy, intradermal IFNγ administration can alter local infections from lepromatous to tuberculoid leprosy, increasing the numbers of CD4+ cells and reducing bacterial numbers in dermal biopsies [23,24]. IL-12 administration has also been shown to cure mice infected with *Leishmania major* [25]. Finkelman et al. [26] modulated the mouse response to *Nippostrongylus braziliensis* from a Th2-dominant response, characterised by IL-3 and IL-4 production, by parenteral administration of IL-12. *In ovo* administration of chicken IFNγ is being considered for protection against a number of avian pathogens, including chicken anaemia virus, and for its adjuvanticity in vaccine formulations [27,28]. 

Intravenous administration of a single large dose of recombinant chicken IFNγ during persistent *S*. Pullorum infection can lead to a reduction in the total number of *S.* Pullorum-infected spleens: 4/18 (22%) spleens positive for *S*. Pullorum in the IFNγ-treated animals and 7/13 (54%) in the untreated controls (*p* < 0.01) [19]. Recombinant chicken IFNγ has also been used to enhance NO production in avian macrophages and reduce the intracellular replication of *S*. Typhimurium and Enteritidis [29]. 

It seems highly unlikely that such an approach would be considered for *S*. Pullorum in susceptible commercial chickens for economic reasons, and because there are other ways to eliminate carrier birds by a test-and-slaughter policy. However, it could have enormous benefits in reducing persistence in the liver and spleen with gall bladder infection in human typhoid carriers if administered during acute infection. Evidence suggests that the typhoid carrier state in India may reach 8% of healthy residents [30]. For exploring the potential of this approach in this important human pathogen, we recommend a combination of in vitro studies involving *S.* Typhi-infected human macrophages co-cultured with T lymphocytes coupled with chemokine analysis of human carriers which could indicate whether this is likely to be a profitable way forward.

## 3. AMR in Pathogenic and Commensal Bacteria

### 3.1. The Problem

Antimicrobial resistance (AMR) is now regarded as one of the major existential threats to human and animal health in our century [https://www.un.org/pga/71/wp-content/uploads/sites/40/2016/09/DGACM_GAEAD_ESCAB-AMR-Draft-Political-Declaration-1616108E.pdf (accessed on 21 June 2022)]. Human deaths attributable to AMR are now estimated to be in excess of 1 million p.a. [1]. One report [31] has argued that AMR-associated mortality could reach 10 million p.a. by 2050, affecting global GDP by 2–3.5%.

*Escherichia coli* is regarded as a major driver of AMR [32] in part because it is a common intestinal commensal and, as such, is exposed to many antibiotics particularly when administered orally. In addition, most AMR in *E. coli* is mediated by self-transmissible plasmids which can be transmitted to other enteric bacteria including several of the opportunistic ESKAPE pathogens [33]. 

Plasmids are transmitted involving the large transfer (*tra*) operon encoding 40 genes, one of which encodes the production of pilin, a protein which coalesces into a filamentous appendage, the sex pilus which attaches to potential recipient bacterial cells. It has long been known that these pili are receptors for two types of bacteriophages. The interaction between plasmids and phage is complex with many plasmids repressing self-transmissibility in order to maintain a degree of transmissibility combined with phage-resistance. Nevertheless, some AMR plasmids have been able to spread globally despite high levels of repression [33] 

National and international governments and institutions [34,35,36,37] have called for a comprehensive strategy, with tighter national regulation, improvements in diagnosis, the identification of new drugs, and the development of completely new and imaginative approaches to tackling the problem [38]. These measures together with improvements in animal husbandry/biosecurity, vaccine use and probiotic development are likely to result in positive, albeit slow, reductions in AMR [39]. However, none of these measures will have an effect on existing infections caused by bacteria which are already resistant. Bacteriophages (viruses that infect bacteria) have been identified specifically as one of several different approaches to combatting AMR [34,35,38]. 

### 3.2. New Approaches to Address the Problem

Lytic bacteriophages have been proposed for treatment of bacterial diseases several times since their discovery in the early years of the 20th century. Poor understanding of their nature and of bacterial pathogenesis, together with poor experimental design and the advent of antibiotics, led to their neglect in the west by the 1950s, although they continued to be used in Eastern bloc countries. Interest was resurrected in the 1980s after some experimental work in animals [40,41] showing that they could be highly effective and more effective than antibiotics [40,42]. This has led to a resurgence of interest in their use but the technology has not yet fulfilled its clinical potential.

Many phages use surface structural components such as lipopolysaccharide as receptors, which limits the range of targeted bacteria. One key negative feature of phage therapy is the development of phage resistance during treatment. This may be reduced by using phages that target surface virulence determinants as receptors so that most phage-resistant mutants will not possess the surface receptor/virulence determinant and thereby be less virulent [40]. 

This strategy is being used by two research groups working on phages that target the sex pili produced by AMR plasmids, such that the majority of phage-resistant mutants have lost their AMR plasmid [43,44,45].

Tectiviruses and Emesvirus (former Leviviruses) have been used in in vitro and in vivo studies to effectively replace AMR strains by their antimicrobial sensitive (AMS) derivatives for plasmid curing in vitro and in vivo using different animal models. A small proportion of bacterial cells in any plasmid-containing culture lose their plasmid as a natural random but infrequent event. Interestingly, this occurs regardless of the ubiquity of toxin-antitoxin (TA) systems, which should prevent its occurrence by killing plasmid-free bacterial cells that occur as a result of daughter cells not inheriting a plasmid copy. In the presence of an antibiotic, these plasmid-free cells would normally die as a result of the TA systems. However, in the absence of antibiotics and the presence of these phages, the plasmid-free cells show greater fitness and may outgrow the remaining plasmid-containing parent cells which are killed by the phage. The presence of male-specific phages thus exerts a strong selection pressure against highly self-transmissible plasmids. The Tectivirus PRD1 has been used against the RP4 plasmid encoding resistance to kanamycin, tetracycline and ampicillin. Similarly, the Emesvirus MS2 has been shown to be able to select for complete replacement of *E. coli* or *S*. Enteritidis harbouring an IncF ampicillin plasmid [45] both in vitro and in the intestine of a group of chickens. In both these sets of studies, the plasmids were highly self-transmissible with >90% of bacterial cells producing the phage receptor. In both cases, relatively short periods of incubation of the AMR bacteria with the phage resulted in the plasmid-cured derivatives becoming dominant. In both cases, a small proportion of phage-resistant mutants retained the plasmid but had ceased to be self-transmissible with mutations in genes in the transfer regions (*traJ* in the case of F [45] and *trbIJ* and *K* in the case of RP4 [43]).

The plasmids studied by both groups were highly self-transmissible with >90% of the bacterial cells possessing the plasmid also producing sex pili and are thus phage-susceptible. In reality, the proportion of cells expressing pili in most clinical strains is much less than 1%, so some means of increasing pilus production through pharmaceutical de-repression would be required for this approach to be used clinically. However, it does illustrate that AMR strains can be replaced by AMS derivatives. Such an approach could extend the clinical life of many antibiotics.

The advantages to this approach are manifold, namely: (i) phages can be found for most incompatibility group plasmids, (ii) the phages kill bacterial cells possessing highly transmissible plasmids, (iii) AMR strains are replaced by the antibiotic-sensitive derivatives, (iv) the phages block conjugation, (v) plasmid-free bacteria multiply faster than the AMR plasmid-containing parent strain, (vi) phage specificity is to the plasmid not the bacteria, enabling targeting of the AMR plasmids in different bacterial taxa.

This approach may also have applicability to reduce AMR where the genes involved are not plasmid-mediated. Chan et al. [46] isolated a Myoviridae phage OMKO1, for which the receptor was the OprM porin from the MexAB and MexXY drug efflux pump system in a multi-drug-resistant strain of *Pseudomonas aeruginosa*. Phage resistant mutants developed following phage activity in vitro which resulted in increased susceptibility to a range of antibiotics, resistance to which was mediated by efflux activity. 

### 3.3. Parasite Viruses and Virus Therapy—A New Approach to Difficult Diseases?

Considerable attention has been given over many decades to the viruses of bacteria (bacteriophages), animals and plants, from the point of view of the mechanism of disease, but also from the point of view of obtaining a better molecular understanding of viruses and microbial genetics. It has been postulated for many years that viruses that infect eukaryotic parasites, including protists, nematodes, cestodes and trematodes, would be discovered. This could lead to progress in understanding the infection biology and epidemiology of these viruses but also, more tentatively, to exploring how they might be exploited for controlling parasite infections, in the same way that bacteriophages may be used. This process began in the early 1970s when virus-like particles began to be observed in the cytoplasm of free-living protists.

The case for novel approaches to addressing some of these parasites has never been stronger. 

Parasite infections of humans and livestock place a huge public health and economic burden on mankind [47,48]. Malaria is responsible for 10% of all childhood under-five deaths in developing countries (http://www.who.int/malaria/en/index.html (accessed on 21 June 2022)) with increasing concerns over drug resistance and with a cost estimated at $12 billion in Africa in lost GDP. Schistosomiasis, also known as bilharzia, infects an estimated 200 million people worldwide (http://www.who.int/mediacentre/factsheets/fs115/en/index.html (accessed on 21 June 2022)), with an estimated cost at >$500 billion p.a. Filarial nematode worms are responsible for disfigurement through elephantiasis and eye infection affecting 120 million people worldwide (http://www.who.int/mediacentre/factsheets/fs102/en/ (accessed on 21 June 2022)) and river blindness (*Onchocerca volvulus*) affects 37 million people in sub-Saharan Africa with particular impact on women (www.apoc.bf/ (accessed on 21 June 2022)). 

In Europe and the rest of the world, helminths, namely nematodes (roundworms), cestodes (tapeworms) and trematodes (flukes) remain major causes of economic loss in livestock rearing with zoonotic infection. A wide range of roundworms affect the intestines, lungs and other organs including the heart, in man, livestock and companion animals. The cestodes include the *Taenia*, *Diphyllobothrium*, *Echinococcus* and *Hymenolepis* tapeworms of man and animals. Trematode (fluke) infections include *Fasciola hepatica*, the liver fluke causing huge losses to cattle and sheep (estimated at <£23 million in the UK alone) and associated human disease and *Clonorchis sinensis* affecting 200 million people at risk in China and the Far East and contributing to liver cancer. Amongst the protozoans, *Giardia lamblia* and *Cryptosporidium* spp. cause an unquantified number of cases of enteritis in man and calves, respectively, with associated economic loss, and with zoonotic transmission of the latter. *Toxoplasma gondii* can cause a fatal and debilitating brain and eye disease in immunocompromised persons or in the developing foetus which occurs as a result of cross-placental transmission during pregnancy. *Trichomonas vaginalis* is a significant cause of venereal disease in many countries. *Eimeria* spp. cause avian coccidiosis resulting in losses to the European broiler industry in excess of EUR 4.8 billion p.a. Live vaccines are available but are expensive and failure can occur [49,50].

The costs of chemotherapy and, in many cases, the associated side effects are considerable and, in the case of malaria, resistance to most available drugs is common. Vaccination has made recent considerable progress for malaria but more generally remains difficult given the complex nature of the parasites.

Viruses have now been found in a wide range of protists including free-living and parasitic amoebae, the flagellates *Trichomonas vaginalis* and *Giardia lamblia*, several *Leishmania* spp. and *Plasmodium vivax* the apicomplexans *Eimeria*, *Cryptosporidium* and parasitic platyhelminths ([49,50], Table 2) and the helminth *Caenorhabditis elegans* [51]. Viruses have not yet been discovered for *P. falciparum*, *Trypanosoma brucei* or *T. cruzi* or the many parasitic helminths. 

With a few exceptions, most of the protist viruses are Totiviruses, small viruses with 2 ORFs and 4–6 kbp. The viruses can be shown to transmit horizontally and are also known to affect the parasite phenotype including alterations in surface proteins and lysis of *Entamoeba histolytica* and *T. vaginalis* and pathological changes in *C. elegans* (Table 2 [52,53,54,55,56,57,58,59,60,61]).

Viruses can already be applied in the field to modulate virulence albeit in fungi. Mycoviruses have been isolated from a number of phytopathogenic fungi and can induce phenotypic changes including hypovirulence, as in *Chryphonectria parasitica*, or increased virulence in *Helminthosporium victoriae*. The cause of Chestnut blight, *Chryphonectria parasitica,* harbours a Chryphovirus CHV1-EP713 which replicates very actively in the fungus and induces hypovirulence and female infertility. Dissemination of hypovirulent strains by artificial introduction induces the same attenuating changes in virulent strains of the fungus with beneficial effects on the severity of tree disease [62].

Viruses, either native or engineered, have also been employed successfully as highly effective and selective therapeutic approaches in China to treat cancer (oncolytic viruses) [63].

We can logically expect that, by analogy with bacteriophages and mycoviruses: (i) viruses will be found in all the major taxa of eukaryotic parasites including the major pathogenic protists and helminths (cestodes, trematodes and nematodes, and it would be interesting to speculate on their involvement in the complex life cycles of some trematode species), (ii) some viruses will have the capacity to alter a virulence phenotype, (iii) some viruses at least will have a virulent, lytic relationship with the parasite host. 

These three points suggest that, by analogy with bacteriophages and mycoviruses, some parasite viruses may be exploitable to control/ameliorate parasite infections of man and animals. The time has perhaps come for exploring in greater detail the infection biology of parasite viruses including their capacity and potential for modulating /moderating the course of parasite viruses [49,50].

## 4. Conclusions

A number of clinical problems remain to be resolved in human and veterinary medicine. We have identified three of these, namely, persistent carrier state that occurs in a proportion of convalescents recovering from typhoid-like *Salmonella* infections in man and livestock, plasmid-mediated antibiotic resistance, which is now seen as an existential global problems for the 21st century and which can only be partially resolved by improvement in registration, reduced use and better diagnosis, and parasite infections, most of which have not yet been studied in depth, which is necessary to entertain any hope of vaccine development. Rather than look again at existing approaches to tackling these problems, we present here novel approaches using existing technologies but adapted to individual problems and propose future research work be directed towards increased development of these approaches and their application.

## Figures and Tables

**Table 1 microorganisms-12-00421-t001:** Infection biological characteristics of currently predominant *Salmonella* serovars [3,4,5,7,8,9,10].

*Salmonella* Serovar	Host Specificity	Effect on Man	Effect on Target Host	Gut Colonizer as Major Feature	Persistent Systemic Infection	Multiplication in Macrophages Central	Nature of Immune Response
*S*. Infantis	Non-specific (mainly poultry—zoonotic)	Enteritis	Virtually no disease	Yes	No	No	Not known—probably Th1-type
*S.* Heidelberg	Non-specific (mainly poultry—zoonotic)	Mainly enteritis	Virtually no disease	Yes	No	No	Not known—probably Th1-type
*S*. Typhimurium	Non-specific—zoonotic	Mainly enteritis	Typhoid in mice. Disease in young chickens and calves	Yes	In Salmonella-resistant mice	Yes, in mouse typhoid	Th1-type
*S*. Enteritidis	Non-specific (mainly poultry—zoonotic)	Mainly enteritis	Typhoid in mice. Disease in young chickens	Yes	Possibly same as Typhimurium	Yes, in mouse typhoid	Th1-type
*S*. Typhi	Man	Typhoid	Typhoid	No	Yes	Yes	Th1 and Th2 depending on host genetics
*S.* Gallinarum	Birds esp. poultry	Rare systemic disease	Typhoid	No	Yes—mainly in Salmonella-resistant chickens	Yes	Anti-inflammatory
*S*. Pullorum	Birds esp. poultry	Rare systemic disease	Systemic disease in young chickens	No	Yes—in Salmonella-susceptible chickens	Yes	Th2-type
*S*. Dublin	Mainly cattle—also mice	Infrequent severe systemic disease	Enteritis and systemic disease cattle, typhoid in mice	In enteritis but not systemic disease	Yes, but unclear	Yes, in systemic disease	Not known
*S*. Abortusovis	Mainly sheep and goats—also mice	Rare systemic disease	Enteritis and systemic disease. Typhoid in mice	In enteritis but not systemic disease	Yes, but unclear	Yes, in systemic disease	Not known

**Table 2 microorganisms-12-00421-t002:** Major classes and characteristics of parasite viruses where this is known.

Parasite	Virus	Nucleic Acid	Effect on Parasite	Reference
**Protists**				
*Leishmania*	Totivirus (Leishmaniavirus), Leishbunyavirus	RNA	Increasing virulence—inflammatory	[52,53]
*Giardia*	Totivirus (Giardiavirus)	RNA	Cell division affected	[54]
*Trichomonas*	Totivirus (Trichomonasvirus)	RNA	Lysis and giant cell formation—increased inflammation	[55,56,57]
*Cryptosporidium*	Partitivirus (Cryspovirus)	RNA (segmented)	Altered fecundity	[58]
*Plasmodium*	Narna-like virus	RNA	Not known	
*Eimeria*	Totivirus, Victorivirus (Eimeriavirus)	RNA	Not known	
*Entamoeba*	Mimivirus	DNA	Lysis	[59]
*Naegleria*	Virus-like particles	Not known	Lysis	[60]
**Selected metazoans**				
*Caenorhabditis*	Related to Nodaviruses	RNA	Cytopathology	[61]
Other nematodes (e.g., *Schistosoma*)	Nothing so far	Not relevant	Not relevant	
Trematodes	Nothing so far	Not relevant	Not relevant	
Cestodes	Nothing so far	Not relevant	Not relevant

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
