# Peer review of "New Approaches to Tackling Intractable Issues in Infectious Disease"

_microorganisms, 2024, doi:10.3390/microorganisms12030421_

Round 1

Reviewer 1 Report

Comments and Suggestions for Authors

The manuscript presents a concise review that covers references from the 1960s to the present, addressing innovative strategies for controlling multi-resistant pathogens in persistent infections.

Comments:

Line 23: replace S. Typhi with Salmonella Typhi

Line 65: Italicize "S" (S. Gallinarum, S. Pullorum)

Line 68: Please, correct: "clearing responses [11]. Studies..."

Lines 72-73: The sentence "and generation of IL-12 and IL-18 early in infection."appears to be missing a reference.

Line 74. Table 1: Add the references used to construct the table. If possible, use more current references. ItalicizeSalmonella; remove the question mark (?) from the sentences "? Possibly same as Typhimurium" and "Th1 and Th2 depending on host genetics ?"

Line 75: In this topic, the role of interferon gamma in controlling the growth of the pathogen, both in the early and late stages of the disease, remained to be explored. See some references:

- Bao S, Beagley KW, France MP, Shen J, Husband AJ. Interferon-gamma plays a critical role in intestinal immunity against Salmonella typhimurium infection. Immunology. 2000 Mar;99(3):464-72. doi: 10.1046/j.1365-2567.2000.00955.x. PMID: 10712678; PMCID: PMC2327174.

- Ingram JP, Brodsky IE, Balachandran S. Interferon-γ in Salmonella pathogenesis: New tricks for an old dog. Cytokine. 2017 Oct;98:27-32. doi: 10.1016/j.cyto.2016.10.009. Epub 2016 Oct 20. PMID: 27773552; PMCID: PMC5398957.

- Maciel BM, Sriranganathan N, Romano CC, dos Santos TF, Teixeira Dias JC, Gross E, Rezende RP. Infection cycle of Salmonella enterica serovar Enteritidis in latent carrier mice. Can J Microbiol. 2012 Dec;58(12):1389-95. doi: 10.1139/cjm-2012-0375. Epub 2012 Nov 27. PMID: 23210996.

- Monack DM, Bouley DM, Falkow S. Salmonella typhimurium persists within macrophages in the mesenteric lymph nodes of chronically infected Nramp1+/+ mice and can be reactivated by IFNgamma neutralization. J Exp Med. 2004 Jan 19;199(2):231-41. doi: 10.1084/jem.20031319. PMID: 14734525; PMCID: PMC2211772.

Lines 80-81: I think the sentence "Persistence occurs despite very high levels of circulating specific IgG [10]." needs to be better explored. Why does this occur?

Line 121: Plese, correct "We have shown [19] that..."

Lines 153-156: Please explain the sentence further: "The interaction between plasmids and phage is complex with many plasmids repressing self-transmissibility in order to maintain an uneasy equilibrium between transmissibility and phage resistance."

Line 164-166: "Bacteriophages (viruses that infect bacteria) have been specifically identified as one of several novel approaches to combating AMR [35, 36, 39]." I don't think it's "novel".

Lines 178-180: Please explore the sentence further: "This may be reduced by using phages that target surface virulence determinants, thus rendering phage-resistant mutants less virulent [41]."

Lines 185 and 209: Please clarify "AMS derivatives."

Lines 188-189: Please explain the sentence further: "Interestingly, this occurs regardless of the ubiquity of Toxin-Antitoxin systems which should prevent its occurrence"

Lines 211-216: number (ii) is repeating. There are (iv) items in total.

Line 256: spp not in italics.

Line 290: Please, correct "mycoviruses:"

Line 304: italicize Salmonella

​

Author Response

Please see attached the comment from the reviewer with my response in italics.

Reviewer 2 Report

Comments and Suggestions for Authors

The title "New Approaches to Tackling Intractable Issues in Infectious Disease" as well as the Abstract suggests a very comprehensive review of different approaches to tackle different issues related to treatment and control of infectious diseases in general . However there is basically one approach very briefly mentioned (immunomodulation) without any real conclusion and one approach explained more in depth (by bacteriophages/viruses) although I find little/no evidence from any clinical trial/real life situation . Is there none?

And only a few specific infections are mentined in the text of the article 

This restricted perspective should be made more explicit and the "approaches" should be clearly highlighted in the Abstract 

Comments on the Quality of English Language

Requires editing 

Author Response

see attached reviewer's comment with my response in italics.
